# Exploring the Influence of a Community-Based Peer-Led Wheelchair Skills Training on Satisfaction with Participation in Children and Adolescents with Cerebral Palsy and Spina Bifida: A Pilot Study

**DOI:** 10.3390/ijerph191911908

**Published:** 2022-09-21

**Authors:** Béatrice Ouellet, Krista L. Best, Deb Wilson, William C. Miller

**Affiliations:** 1Department of Rehabilitation, Faculty of Medicine, Université Laval, Quebec City, QC G1V 0A6, Canada; 2Center for Interdisciplinary Research in Rehabilitation and Social Integration, Centre Intégré Universitaire en Santé et en Services Sociaux de la Capitale-Nationale (CIUSSS-CN), Quebec City, QC G1M 2S8, Canada; 3Seating To Go—Geneva Healthcare, Hamilton 3204, New Zealand; 4Department of Occupational Science & Occupational Therapy, University of British Columbia, Vancouver, BC V6T 2B5, Canada; 5G.F. Strong Rehabilitation Centre—Rehabilitation Research Lab, Vancouver, BC V5Z 2G9, Canada

**Keywords:** children, adolescents, manual wheelchair, power wheelchair, peer-support, wheelchair training, participation

## Abstract

Background: Peer-led approaches improve satisfaction with participation, wheelchair skills and wheelchair use self-efficacy in adults, but the evidence is limited in children. This pilot study aimed to explore the influence of community-based, peer-led, group wheelchair training program (i.e., Seating To Go) on satisfaction with participation (primary outcome), wheelchair skills, and wheelchair use self-efficacy in children and adolescents with cerebral palsy and spina bifida. Methods: A single group pre-post design was used. Invitations were shared online and diffused by clinicians and advocacy and provider groups to recruit a convenience sample of eight pediatric wheelchair users. Participants completed the Seating To Go program in groups that were facilitated by adult wheelchair users. Satisfaction with participation (Wheelchair Outcome Measure-Young People), wheelchair skills (Wheelchair Skills Test), wheelchair use self-efficacy (Wheelchair Use Confidence Scale), and perceived wheelchair skills capacity (Wheelchair Skills Test Questionnaire; proxy rating: parents) were evaluated before and after the Seating To Go program. Descriptive statistics and nonparametric longitudinal data analysis were conducted to explore changes in all outcomes from baseline to post-intervention. Results: Pediatric wheelchair users (ranging in age from 5 to 15 years) and their parents reported statistically significant improvements in satisfaction with participation. The improvements in wheelchair skills and wheelchair confidence were also statistically significant, but not the parents’ perception of their children’s wheelchair skills. Conclusions: A community-based peer-led approach to wheelchair skills training seems promising for improving wheelchair outcomes in pediatric wheelchair users. Further controlled studies with larger samples are warranted.

## 1. Introduction

Approximately 2 of 1000 children are born with cerebral palsy [1,2] and 0.34–0.52 of 1000 with spina bifida worldwide [3,4]. These neurological disorders are associated with mobility limitations that can affect participation in activities that contribute to children’s and adolescents’ development, health, and quality of life [5,6]. Walking difficulties have been identified as a factor restricting social participation and community integration for children and adolescents with cerebral palsy and spina bifida, with school, social, recreational and physical activities being particularly affected [5,7,8,9]. Manual and power wheelchairs can support the mobility needs of children and adolescents who have trouble walking, thus may enhance opportunities for participation in these activities [10,11,12]. However, provision of a wheelchair alone does not ensure safe and independent mobility.

Findings from a cross-sectional study revealed that the majority of pediatric wheelchair users (PWCUs) with cerebral palsy did not acquire basic manual wheelchair skills [13,14]. Moreover, many PWCUs require assistance for manual and power mobility [13,15]. In fact, for indoor skills alone, 72% who used manual wheelchairs and 33% who used power wheelchairs needed mobility assistance from a parent [13]. PWCUs also expressed feeling unsafe when navigating their communities where they face challenges such as uneven terrain and potholes [16]. Data from the National Electronic Surveillance System revealed that an estimated number of 42,538 PWCUs under 19 years of age were treated in United States emergency rooms between 1991 and 2008 after a wheelchair tip or fall [17]. These findings suggest that many PWCUs may have difficulties to use their wheelchairs safely and efficiently for participating in meaningful activities, and may benefit from receiving wheelchair skills training.

Evidence from two meta-analyses confirmed the effectiveness of wheelchair skills training for improving wheelchair skills in various adult populations (e.g., spinal cord injuries, musculoskeletal and neurologic disorders, manual and power wheelchair users) [18,19]. The development of wheelchair skills is associated with increased participation in social, community and physical activities [12,20]. However, evidence of the effectiveness of wheelchair skills training for improving the mobility of PWCUs is limited, with four non-controlled trials supporting manual wheelchair skills training [21,22,23,24], weak evidence for power wheelchair training [25], and limited focus on participation [19,25]

Training is recognized as one of eight essential steps of wheelchair service provision [26]. However, therapists in Canadian rehabilitation centers, provide less than three hours of manual wheelchair skills training, with 18% to 37% reporting no training at all due to lack of time, knowledge and resources [14,27]. In North America, clinicians reported providing power wheelchair training one or two times per year [28]. Moreover, when training is provided, it is focused mainly on basic skills, and not on the advanced skills required for navigating in the community (e.g., ascending or descending ramps, sidewalks) [14,27,29]. In pediatrics, it is difficult for clinicians to respond to evolving mobility needs over time as growth and development evolves rapidly during childhood and transitions into new activities and environments are frequent (e.g., transitions from kindergarten to preschool, elementary and high schools) [21,30]. Moreover, many PWCUs do not receive regular rehabilitation services, and thus may miss training opportunities.

Peer-led training, wherein experienced wheelchair users are trained to teach others wheelchair skills through sharing of experiential knowledge, represents one possible solution to expand the delivery of wheelchair skills training into the community [31,32]. Although peer-led training with minimal support of an healthcare professional was effective for improving wheelchair skills and participation in adults [33,34,35], only two case studies have documented the influence of peers on wheelchair skills training in pediatrics. Through verbal instruction and demonstration of how to execute manual wheelchair skills, a nine-year-old [23] and an adult manual wheelchair user supported by health professionals [24] helped PWCUs improve their manual wheelchair mobility.

In 2013, Seating To Go, a private community-based wheelchair assessment and training service in New-Zealand, implemented an evidence-based peer-led wheelchair skills training program for PWCUs. However, the influence of the program on PWCUs’ mobility and participation has not yet been documented. Given the limited knowledge about peer-led approaches to wheelchair skills training in pediatrics, the primary objective of this pilot study was to explore the influence of the Seating To Go program on satisfaction with participation among PWCUs with cerebral palsy and spina bifida. The secondary objectives were to explore the influence of the program on wheelchair skills capacity, parent perceived wheelchair skills capacity, and wheelchair use self-efficacy. The hypotheses were that there would be a statistically significant improvement in satisfaction with participation after the Seating To Go program and increases in all secondary outcomes.

## 2. Methods

### 2.1. Study Design

A pilot-study using a single group pre-post design was conducted in Hamilton and Auckland, New Zealand. Ethics approval was obtained through the Ethics, Quality Assurance and Safety (Health System Improvement and Innovation) of the Ministry of Health New Zealand (#19/NTB/41). All parents provided informed consent and PWCUs provided assent to participate in the study.

### 2.2. Participants

Eight PWCUs were recruited using a convenience sampling method. Invitations to attend the Seating To Go free training sessions were shared online (e.g., Facebook, company website), distributed by clinicians from Seating To Go, and were diffused by advocacy and provider groups (e.g., Ministry of Education). PMWUs who registered for the Seating To Go program and who fit the inclusion criteria were invited to participate in the study. Children and teenagers ranging from 4 to 15 years of age, with cerebral palsy or spina bifida, who had their own manual and power wheelchair, could roll forward without assistance for at least 10 m and were able to follow two-steps instructions (evaluated by a health professional) were included. PMWUs were excluded if they had a medical condition or were anticipated to receive a medical procedure during the time of the study that contraindicated training (e.g., self or parent-reported recent upper extremity injury or injury) or were attending or planned to attend other wheelchair skills training.

### 2.3. Intervention

The Seating To Go program consisted of two, two-hour sessions of wheelchair skills training facilitated by three adult peer-trainers who had between one and eight years of experience with wheelchair skills training. Peer-trainers taught PWCUs how to perform indoor and community wheelchair skills required for everyday life activities (e.g., roll forward, get over obstacles, ascend inclines) using the motor learning principles and wheelchair skills techniques of the Wheelchair Skills Program (WSP) version 5.0 as a guide [36]. A wheelchair and seating therapist and a wheelchair technician were also present to provide assistance when needed. Their roles were to help with time monitoring, to assist spotters to reinforce safety, and to undertake small adjustments to participants’ wheelchair configuration that may facilitate skill development (e.g., elevating footplates, adjusting inflation pressure in rear wheels or the back support height and angle of the wheelchairs). The sessions were delivered in wheelchair-accessible community locations (e.g., community halls and gyms). Seating To Go skills training was delivered in groups of nine (ratio of one peer-trainer for three PWCUs) for fostering observational learning, which is an effective strategy for enhancing motor learning in children and adolescents with physical disabilities [37,38,39]. The peer-trainers completed a one-day wheelchair skills training workshop with Seating To Go wheelchair and seating therapists (occupational or physiotherapists with the New Zealand Level 2 Wheeled Mobility and Postural Management credential). The workshop included background on the Seating To Go group wheelchair skills training, an information and a practical session on the Wheelchair Skills Training Program (WSTP) motor learning principles, an overview techniques and safety considerations for wheelchair skills training, and a discussion on professional behaviour when working with children and adolescents.

Each skills training sessions occurred as follows: (1) a short introduction; (2) a five-minute review of spotter techniques; (3) a 10-min warm-up activity (e.g., follow the leader game [36]); (4) 60-min practice of several wheelchair skills (e.g., turns while moving forward, gets over obstacles) followed by a short game utilizing the skills; (5) 15-min cool-down activity (e.g., scavenger hunt); (6) a group discussion involving parents and PWCUs on the barriers that may be faced when performing the skills in real life settings and solutions to overcome them. Moreover, a 15-min refreshment break was taken at the middle of the session. All training session activities were entirely led by the peer trainers.

During the 60-min skills practice, the peer-trainers explained the skill and provided a demonstration of the appropriate way to perform it. Multi steps skills, that consist of a sequence of sub-skills, were broken down into small components to facilitate the learning (i.e., motor learning principles: simplification and progression) [36]. Moreover, the peer-trainers helped PWCUs remember the steps of the skills by providing verbal cues (e.g., getting over an obstacle: pop, lean, push). They also observed PWCUs and gave them individualized feedback. To ensure PWCUs’ safety when practicing skills, one of their parents acted as spotter. Parents listened to a video prior to the Seating to Go program demonstrating proper spotting techniques that could prevent accidents and injuries. Any adverse events were recorded.

### 2.4. Data Collection

#### 2.4.1. Sociodemographic and Wheelchair Use Information

Parents completed a sociodemographic and wheelchair use questionnaire on the phone with the Seating To Go wheelchair skills coordinator at baseline (i.e., up to a week before the first training session). The questionnaire included the following information: age, sex, ethnicity, diagnosis, type of wheelchair and driving method, number of hours per day in a wheelchair, contexts in which the wheelchair is used, number of years using a wheelchair and the current wheelchair, previous wheelchair skills training, accidents in the past year and use of other mobility devices.

#### 2.4.2. Outcome Measures

Outcome measures were collected at baseline (i.e., up to two weeks prior to the first session) and post-intervention (i.e., up to 21 days after the last training session) for comparison. All outcomes were collected with young people in person in Seating To Go premises in Auckland or Hamilton by wheelchair and seating therapists, except the subjective wheelchair skills capacity that was measured on the phone with parents by the wheelchair skills coordinator. All testers were employees of Seating To Go or Auckland District Health Board who attended a three-hour workshop on the administration and scoring of the assessments led by study investigators or study (KB, BCM) or study coordinator (DW).

#### Primary Outcome

***Satisfaction with participation.*** The Wheelchair Outcome Measure for Young People (WhOM-YP) was used to evaluate PWCUs’s and parents’ satisfaction with participation in meaningful activities that require the use of a wheelchair [40]. The WhOM-YP, a semi-structured survey, was used to identified two to five participation goals of the child and two to five participation goals of the parent for both indoor and outdoor activities [40]. Perceived importance and satisfaction with current participation of the child in the activity were rated on a 11-point ordinal scale ranging from zero (not at all important or satisfied) to ten (very important or satisfied). Mean weighted scores were calculated by multiplying importance by satisfaction and then dividing by the number of participation goals [41]. Child-reported (i.e., children and adolescents who used manual and/or power wheelchairs and aged 5–17 years) and parent proxy approaches have been validated [40].

#### Secondary Outcomes

***Objective Wheelchair Skills Capacity*.** Children’s wheelchair skills capacity was measured using a modified Wheelchair Skills Test—Version 5.0 (WST) for manual or power wheelchair use [21,42,43]. The WST is a standardized measure of the wheelchair users’ ability to perform 33 (manual) or 25 (power) wheelchair skills required in everyday life [36]. Two modifications were made to the tools for the purpose of this study: (1) the item related to transfer was removed as this skill is not taught in a group setting; (2) the items ascending and descending stairs, which are difficult for most pediatric manual wheelchair users and impossible with a power wheelchair, were removed. Each skill was rated on a four-point scale ranging from zero (i.e., majority of evaluation criteria are incomplete, unsafe, unwilling) to three (i.e., highly proficient), with the score depending on both skill execution and safety. A total capacity score was calculated by dividing the sum of the individual skill score with the total possible score multiplied by 100. The WST has only been validated in pediatric manual wheelchair users (i.e., aged 5–21 years) with spina bifida [44].

***Subjective Wheelchair Skills Capacity*****.** The Wheelchair Skills Test-Questionnaire (WST-Q), a self-report tool, was used to assess parent-perceived manual (33 skills) or power (25 skills) wheelchair skills capacity [43,45]. Scoring was done as per the WST. The validity of the WST-Q reported by parent proxy has been documented [44].

***Wheelchair use Self-efficacy.*** The pediatric versions of the Wheelchair Use Confidence Scales for manual (WheelCon-M) and power (Wheelcon-P) wheelchair users were used to evaluate PWCUs’ belief in their ability to use their wheelchair in a variety of challenging situations [46,47]. The WheelCon-Pediatric is a subjective, self-report assessment tool composed of 33 (manual) and 35 (power) items measuring PWCUs’ current perceived level of confidence in six conceptual areas (i.e., navigating the physical environment, performing activities in a wheelchair, problem solving, advocacy, managing social situations and emotions) [46,47]. Responses are scored on a 11-point ordinal scale, with zero being not confident and ten being completely confident [46,47]. The total confidence score was calculated by dividing the sum of all the items’ scores with the number of items.

### 2.5. Analyses

Descriptive statistics (median, interquartile range (IQR), frequency, percentage) were calculated to describe the sample and to summarize all outcomes. A nonparametric longitudinal data (nparLD) analysis was used to explore differences between baseline and post-intervention for the primary outcome (WhOM) and all secondary outcomes (WST, WheelCon, WST-Q). Given the same overall construct was being measured by each tool, the calculated total scores for WST, WST-Q and WheelCon were merged for manual and power wheelchair outcomes. The nparLD test is robust regarding outliers and for non-parametric distributions of ordinal or non-ordinal data that may change overtime, thus has been recommended for small and variable samples size (e.g., *n* = 5 to 10) [48], The nparLD provides estimates of Relative Treatment Effect (RTE) that are evaluated using a range from 0 to 1 (null hypothesis at 0.05), which are interpreted according to an ANOVA-Type Statistic (ATS). A small effect size corresponds to RTE = 0.56, a medium effect size to RTE = 0.64 and a large effect size to RTE = 0.71 [49]. Intention-to-treat analyses were conducted using R Software (V. 1.4.1103; R Foundation for Statistical Computing, Vienna, Austria) and interpreted using ATS (adjusted for pairwise comparisons) with statistical significance assumed as *p* < 0.05.

## 3. Results

Eight PWCUs (and a parent) completed the study between June and December 2019. Children ranged in age from 5 to 15 years (median (IQR) = 10 (2.5) years), were mostly male (*n* = 7), with diagnoses of spina bifida (*n* = 4) or cerebral palsy (*n* = 4) and used a manual (*n* = 4) or power (*n* = 4) wheelchair. Previous experience using the current wheelchair ranged from 1 month to 6 years. Both mothers (*n* = 7) and fathers (*n* = 1) completed the WST-Q. There were no adverse events during testing or training. Further sociodemographic information is presented in Table 1.

### 3.1. Primary Outcome

The WhOM goals identified by PWCUs and parents are presented in Table 2. Children and parents reported statistically significant improvements from baseline to post-intervention in their satisfaction with participation in meaningful activities, (medium effect; *p* = 0.02; see Figure 1 for individual improvements). Table 3 presents a summary of primary and secondary outcomes.

### 3.2. Secondary Outcomes

Children demonstrated statistically significant differences from baseline to post-intervention in wheelchair skills capacity (WST; small to medium effect; *p* = 0.02) and reported statistically significant improvement in confidence (WheelCon, medium effect; *p* = 0.03). However, parents’ perception of their children’s wheelchair skills capacity did not improve significantly (WST-Q-Parents; small effect; *p* = 0.13). Figure 2, Figure 3 and Figure 4 present participants’ individual improvements from baseline to post-intervention for each secondary outcome.

## 4. Discussion

This was the first study to examine the influence of community-based peer-led wheelchair skills training that combined pediatric manual and power wheelchair users. The results were similar to observations made with adults [34,35], such that satisfaction with participation among PWCUs with cerebral palsy and spina bifida and their parents increased after wheelchair skills training. As the difference between pre- and post-intervention satisfaction scores was statistically significant with a medium to large effect, the results are encouraging to support the ultimate goal of rehabilitation (i.e., to facilitate full participation in meaningful occupations). Similarly, statistically significant small to medium effect sizes reported for wheelchair skills and confidence are promising.

Child- and teenager-selected goals were reported in a systematic review as a key ingredient of interventions in occupational therapy [50]. However, few pediatric rehabilitation interventions focus on participation outcomes [50]. The use of the WhOM-YP ensured the Seating To Go program was family centred by involving PWCUs and parents in the establishment of meaningful goals [40,51]. Collaborative goal-setting, essential for fostering an optimal state of engagement in therapy sessions, may have enhanced PWCUs’ and parents’ readiness and willingness to engage in wheelchair skills training [24,52]. Moreover, shared goals between PWCUs and parents may have enhanced commitment, adherence, sense of ownership and empowerment in the intervention process, thus contributing to improved outcomes [24,52,53].

Most of the PWCUs’ goals were related to community activities (e.g., access a park, get to and from school, play games), whereas the parents’ goals were more focused on the child’s function (e.g., activities of daily living) and participation in domestic tasks (e.g., household chores). This is not surprising given that PWCUs with cerebral palsy and spina bifida have less opportunities to participate in activities outside of the home with similar-aged peers and parents commonly experience a high burden from caring for their child with special needs [54,55,56]. For example, a survey across Europe demonstrated that 33% of children with cerebral palsy played sports more than twice a week and 25% participated in housework compared with, respectively, 66% and 50% for their typically developing peers [55]. Another Canadian survey reported that children and adolescents with cerebral palsy who used wheelchairs experienced more disruptions in participation of self-care, domestic tasks, and recreation activities compared to those who do not need mobility aids [5]. Participation was particularly influenced by the severity of the physical impairment and mobility skills [5,55]. A wheelchair training program, such as Seating To Go, may be a promising strategy for reducing the participation barriers associated with mobility limitations, but further studies should be conducted.

The change in wheelchair skills capacity from baseline to post-intervention (small to medium effect) was larger than parent-perceived improvements. This may be explained by several factors. For example, children and adolescents may be able to perform the skills in training and evaluation contexts, which are more reassuring given the presence of peers and spotters. However, they may not use their newly acquired skills in everyday life situations, which tend to be more challenging and less predictable. Another possible explanation may be that the post-intervention evaluations of the wheelchair skills test questionnaire were completed by the parents within one to 21 days after training, thus the parents may have not yet had the opportunity to observe any improvement in the child’s skills. Moreover, parents of children and adolescents with physical disabilities express challenges gauging the appropriate amount of assistance for ensuring success and safety while encouraging independence [57]. This may be particularly the case when PWCUs receive such condensed training and are rapidly acquiring new mobility skills. Over the course of the two training sessions, parents may not have adapted the level of assistance provided to their child. Although the child may have been able to be more independent, parents may have not had enough time to see the progress. On the other hand, learned helplessness is frequently observed in children and adolescents with physical disabilities, who tend to seek help from their parents rather than trying to perform a wheelchair skill [21]. In Gowran’s wheelchair skills training program, parents observed each training session and were provided with an informational guide describing the techniques for executing MWC skills [24]. They reported being able to support their child’s learning and the use of appropriate techniques at home, while having a higher confidence in their child’s abilities [24]. As such, further studies should consider how to include parents in the training.

PWCUs in this study had a higher relative improvement in wheelchair skills capacity after receiving the Seating To Go program (32%) compared to a group of PWCUs who received group-based wheelchair skills training in an institutional setting (14%) [21]. This difference corresponds to the development of approximately six wheelchair skills. From a clinical perspective, this finding is promising given that the acquisition of one additional wheelchair skill could improve function and participation (e.g., being able to roll across a slide slope facilitates travel from home to a friend’s house) [33]. Although both programs used a condensed approach (i.e., two sessions of wheelchair skills training), Seating To Go comprised shorter sessions (two hours versus four hours). This may reinforce Sawatzky’s hypothesis that longer periods of training may not be best for PWCUs because they do not allow time for rest or consolidation [21]. However, the optimal dose and frequency of wheelchair skills training for PWCUs remains unknown and should be further studied [23,43]. Program facilitation may also explain the large improvements seen in the current study compared to Sawatzky’s [21].

Studies have suggested the potential power of peer and group approaches, respectively for providing mentorship and support for people with disability, including wheelchair users [35,58,59,60]. The power of these approaches may extend to children and adolescents as well. Peers share life experiences and are perceived as a credible source of information [59]. Moreover, the vicarious experience that is provided by watching the peer-trainers and the other participants of the group can strengthen self-efficacy in one’s ability to learn new skills and can influence behaviour changes (e.g., wheelchair use) [35,60]. Gowran et al., reported parent perceived benefits of their child interacting with an adult wheelchair user, such that watching another person successfully perform skills fostered their children’s belief in their own potential and revealed skills that were unimaginable before [24]. The results of the current study support this finding and are in line with observations made among adults [24,33,34,35]. Higher self-efficacy is associated with an increased frequency of use of wheelchair skills and higher life-space mobility [61], thus may facilitate participation in PWCUs. In adults, the sociodemographic characteristics of the peer-trainers (e.g., age, diagnosis) were not perceived as important selection criteria to consider by the participants of a peer-led wheelchair skills training program [60]. On another hand, adults who participated in another wheelchair training program had mitigated opinions on group members’ diversity, mentioning advantages and disadvantages of heterogeneity [35]. In pediatrics, children and adolescents’ preferences regarding learning approaches to wheelchair skills training are unknown. However, in domains outside of wheelchair skills training, there is evidence that children and, especially adolescents, tend to be strongly influenced by similar-aged peers [58,59]. In the current study, the peer-trainers were adults and the groups included PWCUs of a wide age range (i.e., 5–15 years of age). Therefore, future studies may explore children’s and adolescents’ experiences of participation in wheelchair skills training programs to better understand the elements that support positive engagement and learning. Moreover, the role of child peer-trainers may be explored.

### Study Limitations

Although the small sample size of mostly males limits generalizability, subgroup analyses (e.g., improvements in manual and power wheelchairs separately), and exploration of confounding factors (e.g., effect of age), additional knowledge about community-based pediatric wheelchair skills training contributes to the dearth of evidence. Manual and power wheelchairs outcomes were merged for analyses. Although the physical and cognitive demands of power and manual wheelchair skills may vary, all tools have been validated for measuring the same constructs (i.e., satisfaction with participation, wheelchair skills, wheelchair use self-efficacy) [40,41,42,44,45,46,47,62]. Given the study design, results should be interpreted with caution. Although promising results were observed for satisfaction with participation, wheelchair skills capacity and wheelchair use self-efficacy, the effectiveness of the Seating To Go program cannot be determined. Moreover, the WhOM-YP and the WheelCon were measured immediately after training (i.e., one to 21 days post-training). Therefore, PMWUs may not have had the opportunity to experience the task demands of newly learned skills in meaningful activities and everyday life settings. For example, improvements in wheelchair use self-efficacy may have been overestimated as the absence of a spotter, the peer-trainer, and the other PWCUs influence confidence when navigating obstacles in the environment. In addition, the absence of follow-up measures limited the possibility to observe if PWCUs have retained the skills they learned in the program over time. Finally, the measurement properties of the evaluation tools have only been evaluated with small samples or specific populations (e.g., WST and WST-Q with 12 PWCUs with spina bifida; WheelCon-M with 58 Dutch PWCUs with physical disabilities) [40,44,63]. In fact, similar to observations made by Huegel et al. and Daoust et al., there were some challenges in using the WST with PWCUs, the most important being to maintain younger PWCUs’ attention and motivation throughout the assessment of the 33 (manual) or 25 (power) skills [44,64]. Further studies should explore the validity of the measurement tools with broader groups of PWCUs, and adaptations could improve the clinical relevance of the WST in pediatrics [44,64].

## 5. Conclusions

The Seating To Go program had a positive influence on satisfaction with participation, wheelchair skills capacity and wheelchair use confidence in a sample of eight pediatric manual and power wheelchair users. This study contributes to expand knowledge on the use of peer-led approaches for ensuring a continuum of wheelchair skills services from rehabilitation institutions to the community. The next steps may involve conducting a randomized control trial with PWCUs to evaluate the effectiveness of the Seating To Go program for improving satisfaction with participation and other mobility outcomes.

## Figures and Tables

**Figure 1 ijerph-19-11908-f001:**
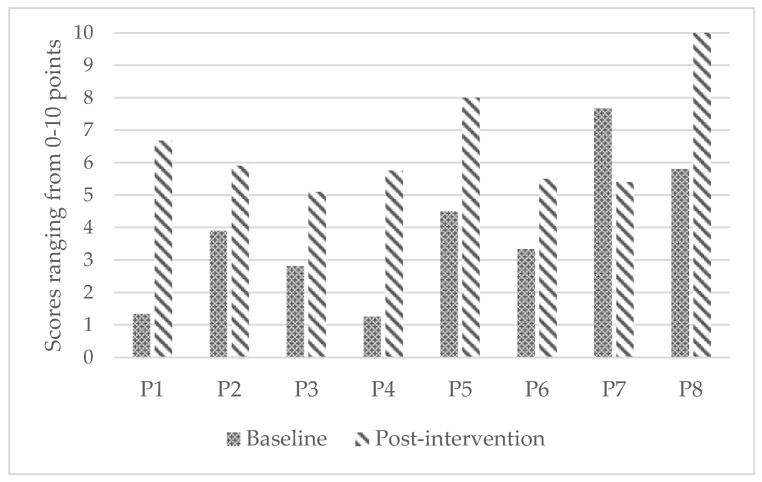
Individual improvements in satisfaction with participation from baseline to post-intervention.

**Figure 2 ijerph-19-11908-f002:**
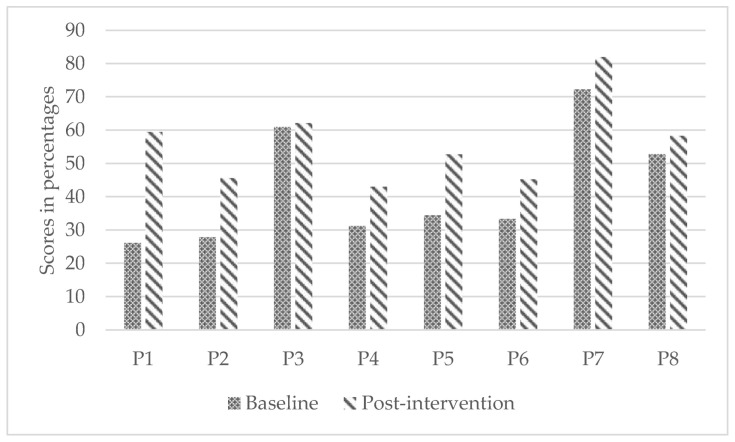
Individual improvements in wheelchair skills capacity from baseline to post-intervention.

**Figure 3 ijerph-19-11908-f003:**
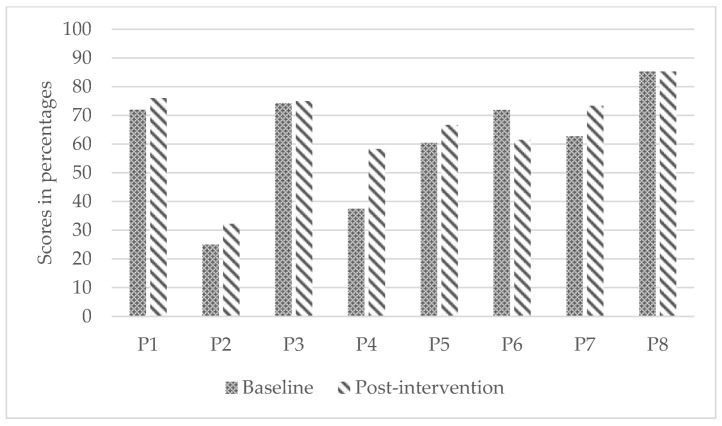
Individual improvements in parent-perceived wheelchair skills capacity from baseline to post-intervention.

**Figure 4 ijerph-19-11908-f004:**
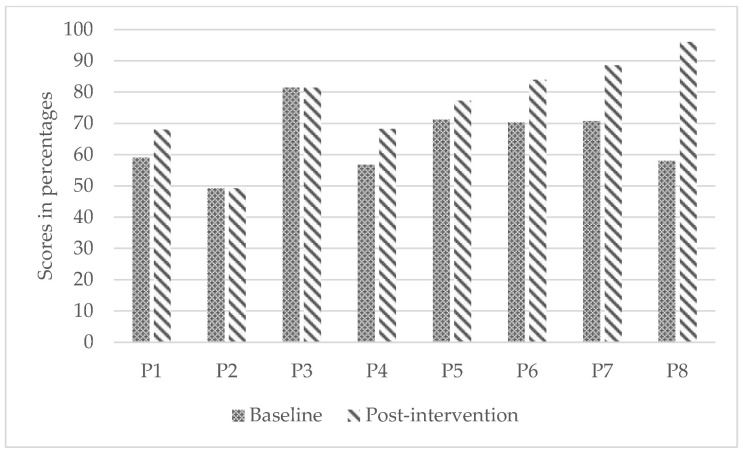
Individual improvements in wheelchair use self-efficacy from baseline to post-intervention.

**Table 1 ijerph-19-11908-t001:** Sociodemographic and wheelchair use information of participants.

*N* = 8	Median (IQR) ^1^	Frequency (%)
Age (years)	10 (2.5)	
Sex (male)		7 (87.5)
Diagnoses:		
Cerebral palsy		4 (50)
Spina Bifida		4 (50)
WC ^2^ Type:		
Manual WC		4 (50)
Power WC		4 (50)
Previous Experience	7.5 (4)	
Daily WC use (hours):		
>8		3 (37.5)
5–8		2 (25)
2–5		2 (25)
<2		1 (12.5)
WC use:		
Everywhere		7 (87.5)
School		1 (12.5)
Accidents in Past Years		2 (25)
Previous WC Training		6 (75)
Used Other Devices		7 (87.5)

^1^ IQR: Inter-quartile range, ^2^ WC = Wheelchair

**Table 2 ijerph-19-11908-t002:** Goals related to wheelchair use as identified by PWCUs and parents.

Children Goals	Parent Goals
Get in/out of the house independently	Clean room
Stay on footpath outside	Transfer
Pop a wheelie at grandparents’ house	Help mother with baking
Chase sisters and cousins	Get a drink from the fridge
Get to/from school	Brush teeth independently
Access a local park	Participate in physical education
Feed rabbit	Get on the floor at Lego club
Wash hands independently	

**Table 3 ijerph-19-11908-t003:** Summary of primary and secondary outcomes, RTE as baseline (T1) and post-intervention (T2), and non-adjusted and adjusted *p*-values.

Outcome Measure	T1 Median (IQR)	T2 Median (IQR)	T1 RTE ^1^	T2 RTE	*p*-Value	Adjusted *p*-Value
WhOM-YP; (out of 10)	3.6 (2.4)	5.8 (1.5)	0.34	0.66	0.006	* 0.02
WST (%)	33.9 (24.5)	55.5 (14.6)	0.38	0.62	0.002	* 0.02
WST-Q—Parent (%)	67.3 (17.9)	70.0 (14.6)	0.45	0.55	0.07	0.13
WheelCon (%)	64.7 (13.1)	79.4 (16.6)	0.38	0.62	0.01	* 0.03

* Statistical significance *p* < 0.05 of adjusted *p*-value, ^1^ RTE: Relative treatment effect, range of 0 to 1 (null hypothesis at 0.05; small effect size = 0.56, medium effect = 0.64 and large effect = 0.71).

## Data Availability

Anonymized data are available from the corresponding author on reasonable request.

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
