# Peer review of "Exploring the Influence of a Community-Based Peer-Led Wheelchair Skills Training on Satisfaction with Participation in Children and Adolescents with Cerebral Palsy and Spina Bifida: A Pilot Study"

_ijerph, 2022, doi:10.3390/ijerph191911908_

Round 1
Reviewer 1 Report
This study focuses on wheelchair training in 5-15 yr old wheelchair users. As outlined by the authors, this is an area that is too often not a focus of clinical practice. Moreover, given the extensive physical and social barriers to full community participation, mobility is a high priority for users and caregivers - even if not currently addressed adequately by clinicians.
This paper was well organized, and very clearly written. I have outlined my major suggestions for revision below.
Please re read and revise overly broad statements based on a single reference.
Example: Pg 1, line 49. This is correct for manual but not for power users.
Another example in Introducation: is the lack of context for the use and safety statements, which although are attempting to make a point, appear over reaching with the current wording.
Another example later in the paper: Observational learning in children is too broad a statement, and is only supported by a single article.
These types of ‘over reaches’ in combination with the conflict of interest (see below) are worrisome in terms of the objectivity of the authors.
Use of “peer led” designation:
Not clear why ‘peer’ is the term used for children being instructed by adults. Yes, both overlap in their use of wheeled mobility but it is not common for children and adults to be considered ‘peers’.
Also it is not clear that with professionals and technicians contributing to the training that this was ‘Led’ by the individuals with the lived experiences using wheeled mobility.
Major conflict of interest:
Devin Wilson is an employee of Seating to Go, which will benefit from positive outcomes, and be challenged by other outcomes.
Application of training and outcomes to pediatric populations:
Not clear that aspects of the peer training were altered to fit a group of such wide age range from 5 to 15 years. Authors acknowledge this towards the end of the paper when commenting on attention span of younger wheeled users to a large number of skills.
These issues need to be acknowledged up front as challenges to applying an adult training to a peds population.
Outcomes:
Need to be more specific about whether outcome measures were appropriate for such a large age range
In table 3 why use mean values with a small sample size and non-parametric statistics?
Would be clarifying to show a graph of Individual subject data pre-and post to see how many of the sample followed the statistical findings. It is not uncommon to have significant individual variably of interest in small sample size studies.
Explanation of the lack of parent perception change is something that could have been assessed. Also an obvious rationale for why parents did not perceive a difference - there was not a difference.
Limitation section: does a good job of alerting the reader to key issues which limited the generalizability of the article. Strong suggestion for the authors to re calibrate their enthusiasm throughout the paper including the title given all the limitations for any first article in an area.
Reviewer 2 Report
The aims of this study were: first to explore the influence of the Seating To Go program on satisfaction with participation among PWCUs with cerebral palsy and spina bifida, and second to explore the influence of the program on wheelchair skills capacity, parent perceived wheelchair skills capacity, and wheelchair use self-efficacy.
Overall, the study is relevant, and the findings are innovative. However, I have a few specific comments
Title: ‘pilot study’ term or something like that should be added in the title given the sample size of the study.
Abstract :
- Please avoid abbreviations in the abstract
- Lines 15-18: ‘wheelchair’ was repeated 4 times, please reformulate
- Line 19: in the method of the abstract, we need to know the sample size of the study. Please add this information.
Introduction:
- the introduction is clear and well structured
- Hypotheses are need for each objective. Please add these information in the manuscript.
Method :
- Line 103: N=8 (4 CP and 4 spina) is very low, could the authors explain in detail how they determine the sample size?
- For intervention and outcomes measure, could the authors add a figure to facilitate the understanding of the procedures
- Line 2019: The authors merged manual and power wheelchairs outcomes please justify this choice and discuss the impact of this choice on the results ‘interpretation.
-
Results: this section is clear.
Discussion: Clear, thank you for the pleasure of reading
Conclusion:
Line 366-369: in view of the small sample size, which does not allow robust clinical conclusions to be drawn, the reviewer suggests that these sentences be put in the conditional tense.
Round 2
Reviewer 2 Report
no additional comments from my side. I thank the authors for the efforts made for the corrections.